# FPGA Implementation of Threshold-Type Binary Memristor and Its Application in Logic Circuit Design

**DOI:** 10.3390/mi12111344

**Published:** 2021-10-31

**Authors:** Liu Yang, Yuqi Wang, Zhiru Wu, Xiaoyuan Wang

**Affiliations:** School of Electronics and Information, Hangzhou Dianzi University, Hangzhou 310018, China; yangliu@hdu.edu.cn (L.Y.); qwaszx20211225@163.com (Y.W.); wu_zhiru@163.com (Z.W.)

**Keywords:** FPGA, memristor, logic gate, adder

## Abstract

In this paper, a memristor model based on FPGA (field programmable gate array) is proposed, by using which the circuit of AND gate and OR gate composed of memristors is built. Combined with the original NOT gate in FPGA, the NAND gate, NOR gate, XOR gate and the XNOR gate are further realized, and then the adder design is completed. Compared with the traditional gate circuit, this model has distinct advantages in size and non-volatility. At the same time, the establishment of this model will add new research methods and tools for memristor simulation research.

## 1. Introduction

In 1971, Professor Chua proposed the fourth passive basic device memristor [1], which is directly used to characterize the relationship between charge and magnetic flux. In 2008, the HP laboratory published the paper “The missing memristor found” in *Nature* and announced the successful implementation of the world’s first memristor physical device [2]. It was found that memristor can perform Boolean logic operations and can completely replace the existing digital logic circuit in theory. In addition, the memristor has a wide application prospect in storage devices, logic circuits, analog circuits and so on.

According to the characteristics of the HP memristor with small ON resistance and large OFF resistance [3], the high and low binary resistance characteristics of the memristor can be applied to digital logic circuits [4,5]. Because the digital logic circuit based on the memristor has a smaller area and lower power consumption, it has been widely studied by researchers in recent years, such as memristor-based material implication (IMPLY) logic circuits, memristor aided logic (MAGIC) circuits and memristor ratio logic (MRL) circuits, etc. In 2010, the HP laboratory first mentioned that the simple circuit composed of memristor and resistor can realize material implementation logic operation (IMP), and then combine FALSE operation to form a logic unit, which can realize the operation of arbitrary Boolean logic function [6]. Then, other circuits based on the memristor by IMP were designed, such as the CMOS-memristor circuit, which can reduce the length of the calculation sequence, or use fewer execution steps and a smaller number of memristors [7,8,9]. Considering the complexity of IMP logic based on the memristor, the MAGIC circuit is proposed, which was simpler and more stable. Subsequently, many improved circuits based on the MAGIC circuit were proposed. By evaluating circuit performance in different ways, it is found that this kind of memristor based circuits with the advantages of reducing circuit area and increasing computing speed [10,11,12,13,14,15]. At the same time, a lot of research has been focused on the MRL circuits composed of memristors and CMOS transistors based on the compatibility of memristor and CMOS transistors. This kind of circuit has fewer devices, less circuit area and power consumption, and also the data density is improved. [16,17,18,19]. In recent years, many digital logic circuits based on memristors and CMOS transistors have been proposed, such as adders [20,21], multipliers [22,23,24], counters [25], decoders [26] and maximum and minimum circuits [14], etc., which play an important role in digital systems.

Therefore, although the fourth kind of basic circuit components, memristor has a great application prospect and research space in the digital logic circuit, all of these memristor based circuits mentioned above focus on circuits design and performance assessment only. Considering the field programmable gate array (FPGA) is a programmable logic array, and it is a good choice to apply memristors to digital circuits, the memristor is modeled in FPGA in this paper. On the one hand, FPGA as a semi-custom circuit in the field of ASIC, not only solves the shortage of custom circuit, but also overcomes the disadvantage of the limited number of original programmable gate circuits. It can program almost infinitely and greatly reduce the cost. On the other hand, a memristor can be applied to digital circuits, and FPGA is a programmable logic array where analog memristor can be directly combined or compared with other gate circuits, without connecting or borrowing other platforms, which reduces the complexity of the transition and comparison steps in the research process and can greatly improve the research efficiency. On this basis, combined with the ON and OFF characteristics of a memristor, the AND gate and OR gate t based on FPGA are realized by the memristor, and the design of the memristor adder is realized too.

The structure of this paper is as follows: firstly, the working principle of the HP memristor is introduced and the circuit model of the threshold memristor is established in FPGA based on this principle; secondly, on the basis of the established model, FPGA AND gate and OR gate circuits based on the memristor are realized. Thirdly, the memristor CMOS and NOR gate, NOR gate, XOR gate and XNOR gate are realized by combining with the NOT gate circuit in FPGA. Finally, the adder is designed by using the logic gates. Compared with traditional CMOS circuits, the memristor-based logic circuit can not only increase the density of the device, but also reduce the power consumption and improve the operation speed of the circuit.

## 2. Working Principle and Threshold Characteristic Analysis of HP Memristor

In 1971, Professor Chua defined the memristor according to the functional relationship between *φ* and *q*: *dφ* = *M*(q)*dq*, where *dφ* = *Vdt*, *dq* = *Idt*. *M* is a variable with the same physical dimension as the resistance, and its value depends on the total amount of charge flowing through the device in the past, so it has a memory function. The memristor has non-volatile characteristics, which are represented by the voltage–current contraction hysteresis characteristics in the circuit. That is, when the input is a periodic signal with zero DC bias, its *V*-*I* characteristic curve shows a “∞” shaped hysteresis curve.

The first memristor in the world was obtained by the HP laboratory through nanotechnology. It was realized by ion doping technology. Specifically, in a very thin piece of TiO_2_, half of it is doped, and there are oxygen deficiencies in the doped side, which is the positive electrode of the HP memristor, and the negative electrode is the undoped side. When a certain voltage level is applied across two electrodes, the oxygen deficiencies in the doped region moves directionally under the action of the electric field, resulting in the width of the doped region and the undoped region changes under the action of the electric field, thus realizing the control of the resistance values of the memristor by the applied electric field. Specifically, when the memristor is positive, the memristor shows a small resistance R_ON_, otherwise it is a large resistance R_OFF_. Figure 1 shows the memristor logic symbol.

As an important 20th century discovery, the HP memristor has attracted wide attention. However, up to now, the HP memristor has not been produced and circulated, and researchers cannot obtain the real object, which makes the simulation research of memristor more important. Therefore, the implementation of the memristor model on FPGA with many resources will be of great significance to the further study of memristors.

## 3. Field Programmable Gate Array (FPGA) Implementation of the Threshold Memristor

According to the second part of the HP memristor threshold characteristics, this paper establishes the corresponding memristor model in FPGA, as shown in Figure 2. In order to highlight the characteristics of the memristor as a bipolar memory device in Quartus II as show in Figure 1, two input terminals InP and InN need to be designed in this model; concretely, InP represents the positive memristor input electrode while InN represents the negative memristor input electrode. One output terminal R_out_ is needed to be established to show the corresponding value of the memristor. The 8-bit binary numbers InP [7..0], InN [7..0] and R_out_ [7..0] are used to represent the forward input voltage, reverse input voltage, and the memristance of the memristor at each voltage, respectively. The data in this design are set to be 8-bit binary number, to ensure the needs of the subsequent computing module, because fewer bits are possible to make overflow during calculation.

According to the analysis in the previous section, when *V*_InP_ > *V*_InN_, the memristor is in the forward conducting state, and the memristor’s value is R_ON_, then R_out_ [7..0] = 00000000; on the contrary, R_out_ [7..0] = 11111111, which means that the memristor resistance value is R_OFF_; when *V*_InP_ = *V*_InN_, R_out_ [7..0] = 00001111, which means the memristance does not change.

The simulation results of the threshold memristor model based on Verilog HDL are shown in Figure 3. When InP = InN = 00000000 or InP = InN =11111111, the result is R_out_ = 00001111, which indicate that the memristance does not change when the two input voltage are same. When InP = 00,000,000 and InN = 1111111, R_out_ = 11111111, that is, under the reverse voltage, the memristance value is R_OFF_. Conversely, when InP = 1,111,111 and InN = 00000000, R_out_ = 00000000, the memristor change to R_ON_. It can be seen that the model meets the working characteristics of the threshold memristor and meets the above design requirements.

Table 1 shows two different ways of constructing an AND gate and an OR gate with the traditional CMOS method and the threshold memristor model based on FPGA we proposed in this paper. From Table 1, we can see that the traditional CMOS based AND gates and OR gates are composed of three pairs of CMOS. However, for the gates based on the threshold memristor model of FPGA we only need two memristors and one computing module, respectively. Compared with traditional CMOS-based gates, the number of components of the proposed circuit is significantly reduced, and the memristor itself has advantages in both size and power consumption, so the corresponding gates composed of memristors are smaller in size and consume less power. Besides, because memristors are compatible with the CMOS technology, so the memristor based compound logic circuits such as NAND gate and NOR gate can be obtained by using CMOS NOT gates, and then combined logic circuits, such as adders, decoders and encoders can be realized too.

## 4. FPGA Implementation of Memristor Basic Logic Gate Circuit

### 4.1. Design of AND Gate and OR Gate Based on Memristor

In order to realize the memristor based AND gate, the positive terminals of the two memristors need to be connected and led out as the output terminals of the AND gates, and the two negative ends are, respectively, used as the input signal terminals [19], as shown in Figure 4a. When both input terminals are at a high level, the output is also at a high level; when both input terminals are low, the output is also low. When the input at both ends is different, the corresponding output of the AND gate is calculated according to the principle of voltage dividing. For example, when *V*_AND1_ is at high level, *V*_H_ = V_DD_, and *V*_AND2_ is at low level, *V*_L_ = 0 V, the current in the circuit will flow from high voltage level to low voltage level. At this time, the resistance of MR_1_ is R_OFF_, and the resistance of MR_2_ is R_ON_. Since R_OFF_ >> R_ON_, the output of the AND gate is *V*_AND_O_ and is determined by the voltage divided by the two memristors, as shown in Equation (1). Because of the symmetry of the circuit, the same result is obtained when the input level is switched.
(1)$$V_{\mathrm{AND}\text{\_}O}=\frac{R_{\mathrm{ON}}}{R_{\mathrm{OFF}}+R_{\mathrm{ON}}}\times (V\mathrm{DD}-0)\approx 0$$

The OR gate design based on the memristor is similar to the AND gate mentioned above, but the difference is that the negative ends of the two memristors need to be connected and led out as the output terminals and the two positive terminals as the input terminals, as shown in Figure 4b. The working principle of the two gates is similar, and the voltage dividing principle is used to complete the calculation. The current flow direction in the circuit is from high voltage level to low voltage level. In the case of *V*_OR1_ = *V*_H_ = V_DD_, *V*_OR2_ = *V*_L_ = 0 V, according to the working characteristics of the memristor, MR_1_ shows low resistance R_ON_, MR_2_ presents high resistance R_OFF_, and the voltage distribution of two memristors determines the output *V*_OR_O_. The value of *V*_OR_O_ is shown in Equation (2). The circuit model has a symmetrical structure, so the same result is obtained when the input conditions are exchanged.
(2)$$V_{\mathrm{OR}\text{\_}O}=\frac{R_{\mathrm{OFF}}}{R_{\mathrm{OFF}}+R_{\mathrm{ON}}}\times (V_{\mathrm{DD}}-0)\approx V_{\mathrm{DD}}$$

### 4.2. Implementation of AND Gate and OR Gate Based on Memristor in FPGA

From the above analysis, to realize the memristor based AND gate and OR gate circuit through FPGA, the two memristors need to be designed in series. Therefore, it is necessary to design a basic operation module to simulate the operation in the actual circuit, as shown in Figure 5. Among them, the input terminals *R*_1_ and *R*_2_ are, respectively, connected with the output terminals of the two memristors in series, that is, the memristor values *R*_out1_ and *R*_out2_ of the two memristors; the input terminals *V*_1_ and *V*_2_ are respectively connected with the two input terminals of the series circuit; the output of the module is the voltage value *V*_OUT_ at the two memristors in series. The module can distinguish whether each memristor is currently in forward bias or reverse bias by the value of *V*_1_ and *V*_2_, so as to obtain the memristor values of the two memristors under the input voltage, and assign the values to *R*_1_ and *R*_2_, and calculate according to Equation (3) to obtain the corresponding output value.
(3)$$\begin{array}{l} V_{\mathrm{OUT}}=\frac{R_{2}}{R_{1}+R_{2}}\times (V_{1}-V_{2})+V_{2}\begin{array}{cc} & \left(V_{1}\geq V_{2}\right) \end{array}\\ V_{\mathrm{OUT}}=\frac{R_{1}}{R_{1}+R_{2}}\times (V_{2}-V_{1})+V_{1}\begin{array}{cc} & \left(V_{1}<V_{2}\right) \end{array} \end{array}$$

In order to verify the correctness of the design, the threshold memristor designed in the previous section and the series operation module established in this section can be used in Quartus II, and the simulation circuit of the AND gate is shown in Figure 6. The negative inputs of the two memristors are used as the input terminals of the whole AND gate, and the *R*_out_ outputs the memristor state according to the level of the input terminal. Then, the output of the whole AND gate is further obtained after the operation of the operation module according to Equation (3). The simulation results are shown in Figure 7. When both *V*_AND1_ and *V*_AND2_ are high level “11111111” or both are low level “00000000”, the output *V*_AND_O_ is the same as the input. When *V*_AND1_ and *V*_AND2_ input different values, the output result represents the low level of 00000000. Therefore, the AND gate circuit designed in this paper is effective. Figure 8 shows the module diagram of the AND gate logic circuit after encapsulation.

Similarly, in order to verify the effectiveness of OR gate circuit, the simulation circuit of OR gate is established as shown in Figure 9. The simulation results of Quartus II are shown in Figure 10, and Figure 11 is the module diagram after encapsulation of the OR gate logic circuit. It can be seen from the waveform simulation diagram of the OR gate in Figure 10 that when the input *V*_OR1_ and *V*_OR2_ are both high or both low, the voltage across the memristor is the same, and the output is the same as the input. When *V*_OR1_ and *V*_OR2_ are different input values, the output result is high level. Therefore, the OR gate designed in this paper is effective.

### 4.3. Design of FPGA Combinational Logic Gate Based on Memristor

In order to realize the memristor NAND gate and NOR gate in FPGA, this paper realizes the memristor NAND and NOR gates by connecting the NOT gate in FPGA after the AND gate and OR gate, as shown in Figure 12.

#### 4.3.1. XNOR Gate Based on Memristor in FPGA

Based on the relationship between NOR and XNOR logic gates as shown in Equation (4), the XNOR circuit based on the memristor in FPGA as shown in Figure 13 can be obtained, and the encapsulated module is shown in Figure 14.
(4)$$v_{\mathrm{XNOR}1}⊙v_{\mathrm{XNOR}2}=(((v_{\mathrm{XNOR}1}+v_{\mathrm{XNOR}2}{)}^{\prime }+v_{\mathrm{XNOR}1}{)}^{\prime }+((v_{\mathrm{XNOR}1}+v_{\mathrm{XNOR}2}{)}^{\prime }+v_{\mathrm{XNOR}2}{)}^{\prime }{)}^{\prime }$$

In order to verify the functionality of the module, circuit simulation is carried out in Quartus II software, and the simulation waveform is shown in Figure 15. When the input signals *v*_XNOR1_ and *v*_XNOR2_ are both at high level or both at low level, the output result *v*_XNOR_O_ is at high level. When the input signals are different, the output result is at low level. It can be seen that the above results fully confirm with the logic function of the XNOR gate.

#### 4.3.2. XOR Gate of FPGA Based on Memristor

The FPGA XOR gate based on the memristor can be composed of the memristor AND gate, OR gate and NAND gate, and can be realized according to Equation (5). The specific circuit is shown in Figure 16, and the encapsulated model is shown in Figure 17.
(5)$$v_{\mathrm{XOR}1}⊕v_{\mathrm{XOR}2}=(v_{\mathrm{XOR}1}+v_{\mathrm{XOR}2})(v_{\mathrm{XOR}1}\cdot v_{\mathrm{XOR}2}{)}^{\prime }$$

Through the simulation experiment of the above circuit, the experimental results shown in Figure 18 were obtained. It can be seen that these results are just opposite to the XNOR gate based on the memristor, that is, when the input signals *v*_XOR1_ and *v*_XOR2_ are the same, the output of *v*_XOR_O_ is at a low level, and the result is at a high level when the input signals are different. This exactly conforms to the logic function of the XOR gate, and also satisfies the rule that XNOR and XOR complement each other.

## 5. Design of Adder Based on Memristor

In a modern computer or digital signal processing system, the operation of data is inseparable from arithmetic logic components, which can perform the logic operation, shift or command call. As one of the most important devices, the adder can complete the addition operation between two numbers, which is the basic unit of the arithmetic unit. In the hardware implementation of various digital systems, the power consumption, running speed and size of adders directly affect the design and implementation of digital systems. Therefore, the design of high-speed and low-power adder circuit is of great significance to a digital system. A memristor has the advantages of fast operation, small size, and low power consumption. The application of the memristor in an adder circuit can greatly improve the performance of the adder. In this section, half adder and full adder are implemented in FPGA by using the logic gates based on memristor.

### 5.1. Half Adder Design

The half adder is the simplest addition circuit. It adds two 1-bit binary numbers A and B to produce sum S and carry CO. The output expression of the half adder is shown in Equation (6), that is, it is composed of an XOR gate and an AND gate. In this paper, a half adder circuit is realized using a memristor-based XOR gate and AND gate shown in Figure 19. Since the carry from low bit is not considered in addition here, the circuit shown in Figure 19 can only be called a half adder, and the encapsulated module is shown in Figure 20.
(6)$$\begin{array}{l} S=A⊕B\\ CO=AB \end{array}$$

Through the establishment of a simulation circuit in Quartus II, the half adder function is tested, and the simulation results as shown in Figure 21 are obtained. In the figure, 000,000,000 is equivalent to 1-bit binary data “0”, 11,111,111 is equivalent to 1-bit binary data “1”. When two addends *A* and *B* are “0”, then *S* and carry *CO* are both “0”; when one of the two addends *A* and *B* is “0” and the other is “1”, carry *CO* is “0”; when two addends *A* and *B* are “0”, carry *CO* is “0” When *A* and *B* are “1”, then *S* and carry *CO* are “1”, the design conforms to the logic rules of a half adder.

### 5.2. Full Adder Design

A logic circuit that adding two binary numbers and considering carry from the low bit to the high bit is called the full adder. The full adder not only considers the addition of two binary numbers, but also takes the carry from the low bit into the addition operation. The expression of the full adder is shown in Equation (7), where *A* and *B* are addends, *C* is the carry digit of the low order, *S* is the standard sum value, and *CO* is the carry forward to the high order.
(7)$$\begin{array}{l} S=A⊕B⊕C\\ CO=(A⊕B)C+AB \end{array}$$

The full adder circuit can be constructed with the memristor-based AND gate, OR gate and XOR gate according to Equation (7), as shown in Figure 22. It can also be composed of two half adders and one OR gate. Figure 23 shows a full adder circuit composed of memristor-based half adders.

Figure 24 shows the simulation circuit results of a full adder based on a memristor built in Quartus II software. Similar to the half adder, 00,000,000 is equivalent to 1-bit binary data “0”, and 11,111,111 is equivalent to 1-bit binary data “1”. When *A*, *B* and *C* of the full adder are all “0”, the standard *S* and carry *CO* of the output end are all “0”; when any one of *A*, *B* and *C* is “1” and the other two are “0”, then *S* is “1” and carry *CO* is “0”; when any one of *A*, *B* and *C* is “0” and the other two are “1”,then *S* is “0” and carry *CO* is “1”; when *A*, *B* and *C* are all “1”, then *S* is “0”, carry *CO* is “1”; when *A*, *B* and *C* are “1”, then *S* is “0”, and carry *CO* is “0” when *A*, *B* and *C* are all “1”, then *S* is “1”, and *CO* is “1”. The design conforms to the logic rules of a full adder.

## 6. Conclusions

In this paper, the circuit model of a threshold memristor is established in FPGA, and the AND gate and OR gate circuits based on the memristor are realized by using this model. Combined with the NOT gate circuit in FPGA, the NAND gate, NOR gate, XOR gate and the XNOR gate based on the memristor are further realized, and the adder is designed based on the above gate circuits. Logic devices are widely used in digital systems. With the development of modern digital technology towards small size, low power consumption and high speed, compared with traditional CMOS circuits, memristor-based logic circuits can not only increase the density of devices, but also reduce the power consumption of circuits, and improve the operation speed of circuits. The emergence of such devices will inevitably bring about great changes in digital technology.

## Figures and Tables

**Figure 1 micromachines-12-01344-f001:**
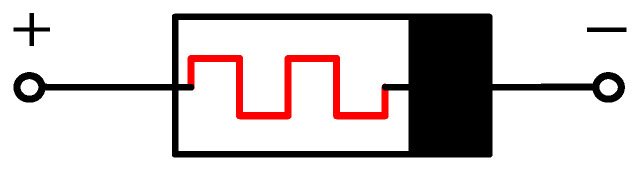
The memristor symbol.

**Figure 2 micromachines-12-01344-f002:**
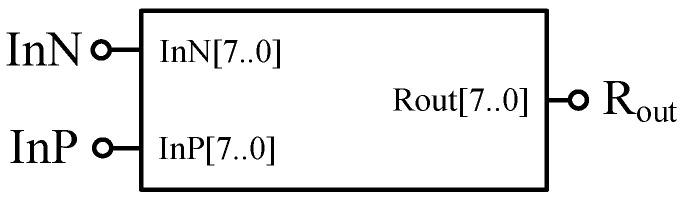
The threshold memristor model based on Verilog HDL.

**Figure 3 micromachines-12-01344-f003:**
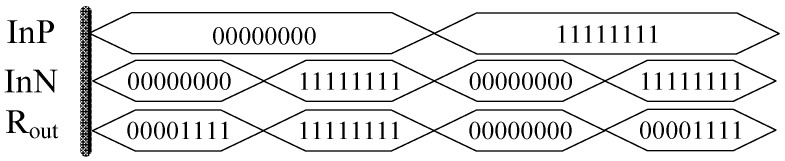
Simulation waveform of the memristor model.

**Figure 4 micromachines-12-01344-f004:**
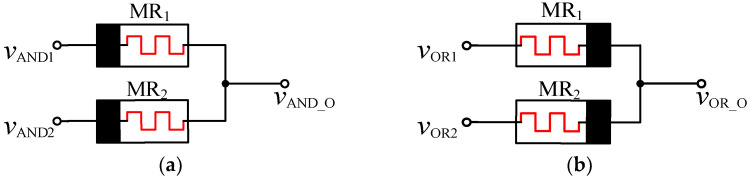
Models of AND and OR gates for the memristor: (**a**) memristor AND gate; (**b**) memristor OR gate.

**Figure 5 micromachines-12-01344-f005:**
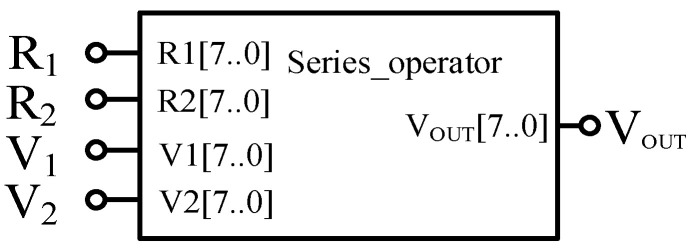
Serial module based on Verilog HDL.

**Figure 6 micromachines-12-01344-f006:**
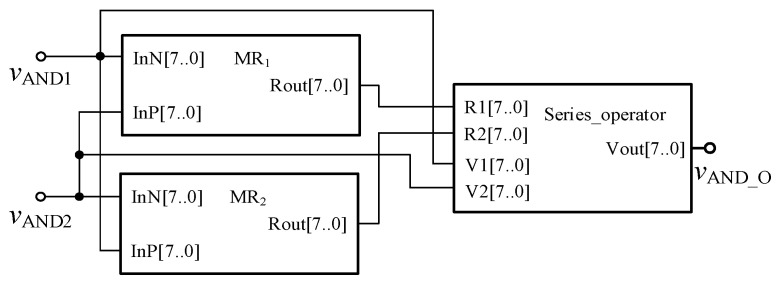
AND gate circuit based on memristor.

**Figure 7 micromachines-12-01344-f007:**
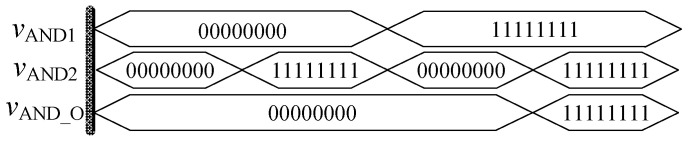
Waveform of memristor AND gate circuit.

**Figure 8 micromachines-12-01344-f008:**
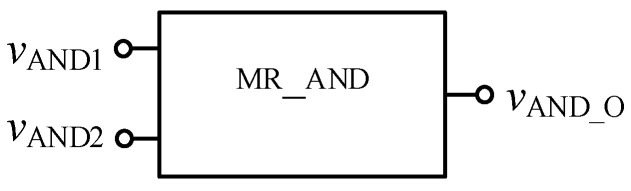
AND gate module after encapsulation.

**Figure 9 micromachines-12-01344-f009:**
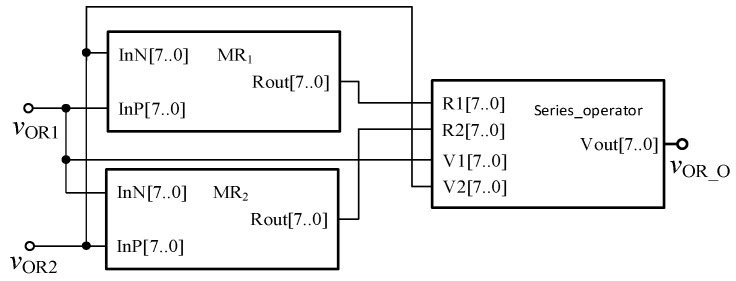
The OR gate circuit based on memristor.

**Figure 10 micromachines-12-01344-f010:**
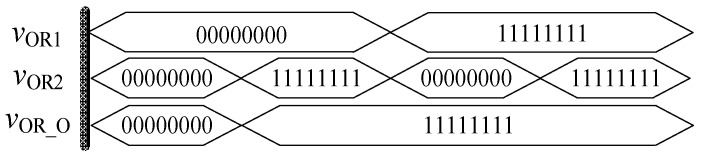
Waveform of the memristor OR gate circuit.

**Figure 11 micromachines-12-01344-f011:**
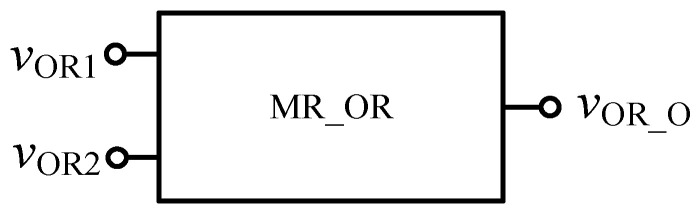
OR gate module after encapsulation.

**Figure 12 micromachines-12-01344-f012:**
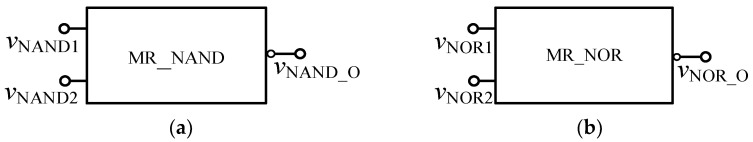
Memristor NAND and memristor NOR gate: (**a**) encapsulated NAND gate module; (**b**) encapsulated NOR gate module.

**Figure 13 micromachines-12-01344-f013:**
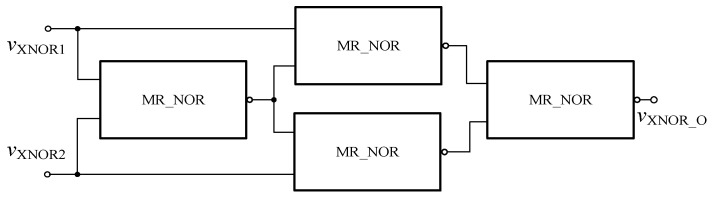
Memristor XNOR gate based on Verilog HDL.

**Figure 14 micromachines-12-01344-f014:**
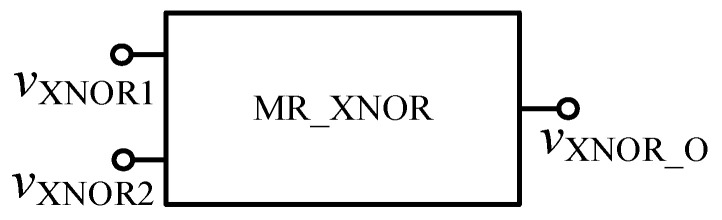
The XNOR gate after encapsulation.

**Figure 15 micromachines-12-01344-f015:**
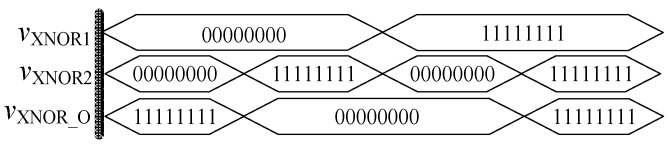
Waveform of the memristor XNOR gate.

**Figure 16 micromachines-12-01344-f016:**
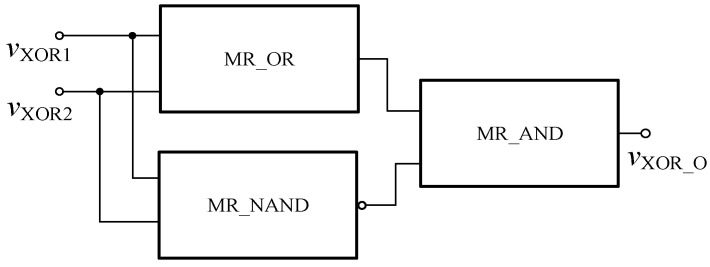
Memristor XOR gate based on Verilog HDL.

**Figure 17 micromachines-12-01344-f017:**
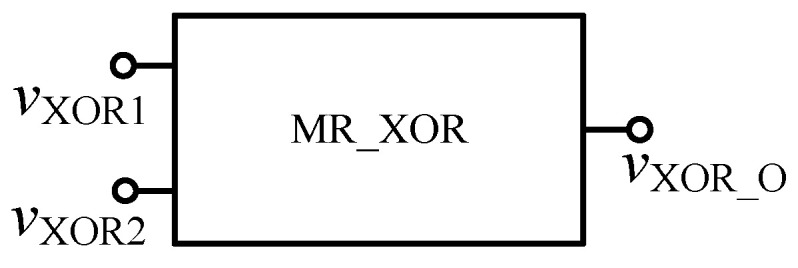
The memristor XOR gate after encapsulation.

**Figure 18 micromachines-12-01344-f018:**
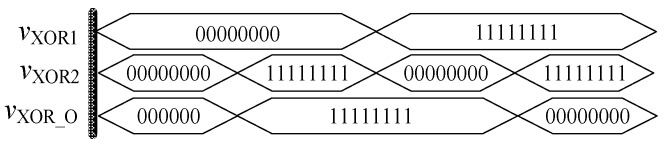
Waveform of the memristor XOR gate.

**Figure 19 micromachines-12-01344-f019:**
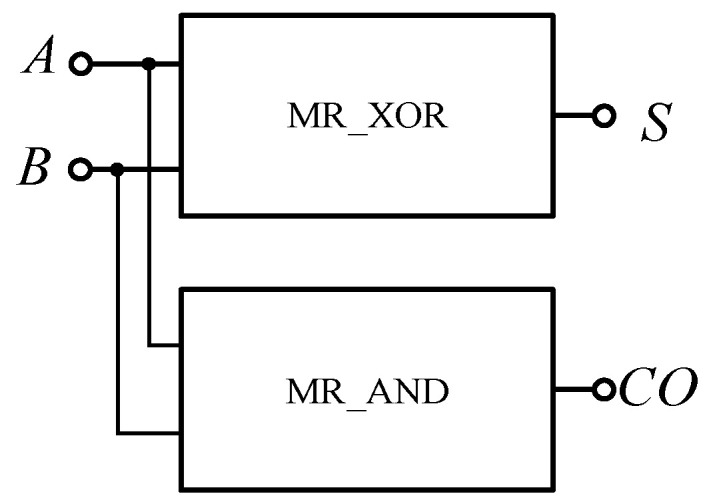
Half adder based on memristor.

**Figure 20 micromachines-12-01344-f020:**
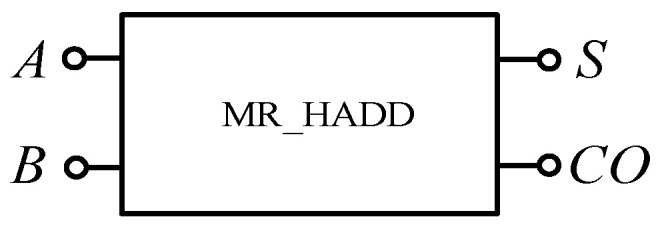
The half adder after encapsulation.

**Figure 21 micromachines-12-01344-f021:**
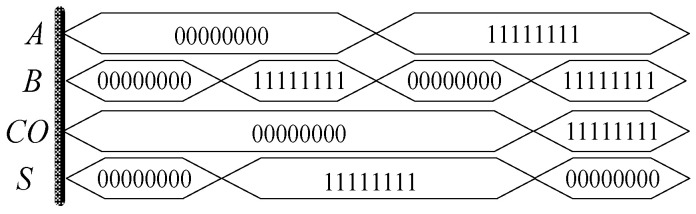
Waveform of the half adder based on memristor.

**Figure 22 micromachines-12-01344-f022:**
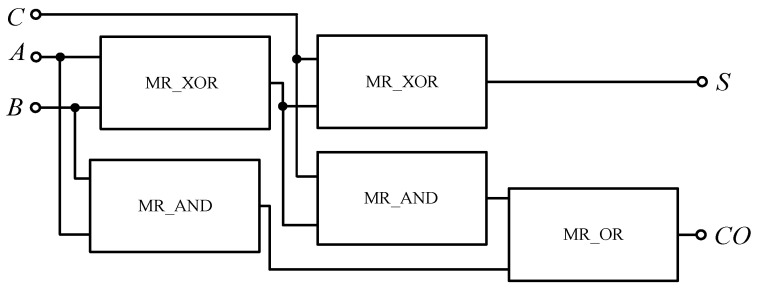
Full adder circuit based on memristor XOR gate, AND gate and OR gate.

**Figure 23 micromachines-12-01344-f023:**
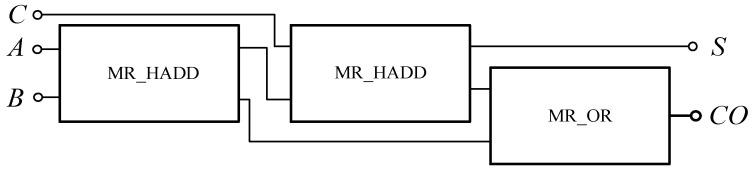
Full adder circuit based on memristor half adders.

**Figure 24 micromachines-12-01344-f024:**
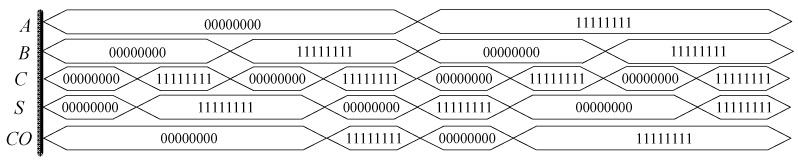
Waveform of the full adder based on memristor.

**Table 1 micromachines-12-01344-t001:** Comparison between gates based on traditional CMOS gates and threshold memristor gates built in FPGA.

	Traditional CMOS Gates	FPGA Model of Threshold Memristor Gates
Circuit Diagram	Logic Symbol	Circuit Diagram	Encapsulated Module
AND gate	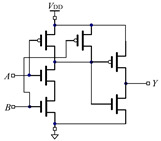	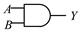	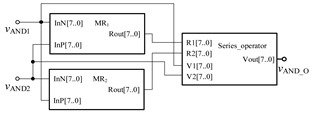	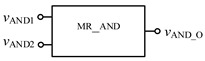
OR gate	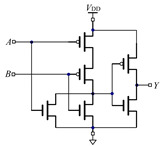	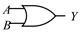	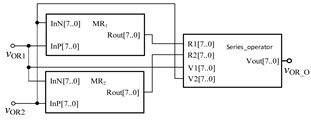	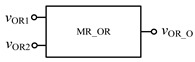

## Data Availability

No new data were created or analyzed in this study. Data sharing is not applicable to this article.

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
