# Peer review of "FPGA Implementation of Threshold-Type Binary Memristor and Its Application in Logic Circuit Design"

_micromachines, 2021, doi:10.3390/mi12111344_

Round 1
Reviewer 1 Report
This manuscript “FPGA Implementation of Threshold Type Binary Memristor and its Application in Logic Circuit Design” by Yang et al. demonstrate the circuit model of threshold memristor based on FPGA. In this study, the author realized the NOT gate in FPGA, and NAND, NOR, XOR, XNOR gate to compared with the traditional gate circuit. I hope the following comments help the authors to improve the manuscript during the revision before considering for publication.
- The introduction part should re-write for more clearly the motivation of this work. The important of this study should be compared with the previously reports.
- The memristor based logic circuit can increase the density, reduce power consumption, and improve the operation speed of the circuit compared to the traditions CMOS circuits. These information’s need to be demonstrated and provided clearly or added the reference.
- The model of logic gate-based FPGA are shown clearly however it is hard to make a comparison with the traditional technology. The author should make a highlight in either table or figure.
Author Response
We wish to thank the Editors and Reviewers for their thorough reviews and constructive comments on our manuscript. We have made careful changes to the text according to the Comments as below:
- We have re-written the introduction. In this version, we have summarized the merits in logic circuits based on memristors, and we found that all of these memristor based circuits are focus on the circuits design and performance assessment. As a basic tool for the research in digital logic circuits, there is no memristor model established in FPGA before, so it is necessary to build a basic memristor module in FPGA to study its possible circuit’s application with the other basic gates in logic circuit. All of the revised parts have been yellowed in the new version.
- The references [7-9,11,13-15,17-20] have been added, to demonstrate the advantage of memristor based circuits in improving operation speed, reducing power consumption and increasing density etc.
- The comparison between the model of logic gate-based FPGA and the traditional CMOS gates has been shown in Table1.

Reviewer 2 Report
The FPGA-based design of memristor logic gates is explained in this paper, with specific implementations (HDL) of the usual logic functions, and binary adders. The authors demonstrate that their FPGA models are functionally correct.
There are a few minor changes that I would suggest in this manuscript:
1) For completeness, the physical operating principle of the memristor should be explained too., particularly concerning the hysteresis of the device.
2) Fig. 2 and 3 and the beginning of Section 3 are the key contributions of the paper. Please, explain the FPGA model with further detail.
3) The manuscript needs a thorough review of the English writing.
Author Response
We wish to thank the Editors and Reviewers for their thorough reviews and constructive comments on our manuscript. We have made careful changes to the text according to the Comments as below:
Response:
- The physical operating principle of the memristor have been added as follow at the beginning of the second section:
In 1971, Professor Chua defined the memristor according to the functional relationship between φ and q: dφ = M(q)dq, where dφ = Vdt, dq = Idt. M is a variable with the same physical dimension as the resistance, and its value depends on the total amount of charge flowing through the device in the past, so it has a memory function. The memristor has non-volatile characteristics, which is represented by the voltage-current contraction hysteresis characteristics in the circuit. That is, when the input is a periodic signal with zero DC bias, its V-I characteristic curve shows a "∞" shaped hysteresis curve.
- To explain the FPGA model and Figs. 2 and 3 more clearly, we have added the following content in the first paragraph in part 3, and rewrite the explanations of these two figures in this version as below:
In order to highlight the characteristics of memristor as a bipolar memory device in Quartus II as show in Figure 1, two input terminals InP and InN are need to be designed in this model, concretely, InP represents the positive memristor input electrode while InN represents the negative memristor input electrode, and one output terminal Rout is needed to established to show the corresponding value of the memristor. The 8-bit binary numbers InP[7..0], InN[7..0] and Rout[7..0] are used to represent the forward input voltage, reverse input voltage, and the memristance of the memristor at each voltage respectively. The data in this design are set to be 8-bit binary number, to ensure the needs of the subsequent computing module, because less bits is possible to make overflow during calculation.
When InP = InN =00000000 or InP = InN =11111111, the result is Rout = 00001111, which indicate that the memristance does not change when the two input voltage are same. When InP = 00000000 and InN = 1111111, Rout = 11111111, that is, under the reverse voltage, the memristance value is ROFF. Conversely, when InP = 1111111 and InN = 00000000, Rout = 00000000, the memristor change to RON.
- We have checked the English of the whole paper, corrected the grammatical errors throughout the paper.

Round 2
Reviewer 1 Report
The revised version appears of clearly and well organize for FPGA implementation of binary memristor in logic circuit design, which has obvious advantages compared to traditional gate circuit, therefore, I recommended to publish this work.
Reviewer 2 Report
The changes made to the manuscript improve substantially the quality of the manuscript, now it's more self-contained and comprehensive. However, the English language and style still need a significant enhancement, for many syntax and grammar errors remain in the text. A thorough and careful revision of the whole text is recommended.
This manuscript is a resubmission of an earlier submission. The following is a list of the peer review reports and author responses from that submission.